# Optimising hydroxyl airglow retrievals from long-slit astronomical spectroscopic observations

Christoph Franzen[1,2], Robert Edward Hibbins[1,2], Patrick Joseph Espy[1,2], Anlaug Amanda Djupvik[3]

[1] Norwegian University of Science and Technology (NTNU), Trondheim, 7491, Norway.
[2] Birkeland Centre for Space Science (BCSS), Norway.
[3] Nordic Optical Telescope, E-38700 Santa Cruz De La Palma, Spain

*Correspondence to:* Christoph Franzen (Christoph.franzen@ntnu.no)

**Abstract.** Astronomical spectroscopic observations from ground-based telescopes contain background emission lines from the terrestrial atmosphere's airglow. In the near infrared, this background is composed mainly of emission from Meinel bands of hydroxyl (OH), which is produced in highly excited vibrational states by reduction of ozone near 90 km. This emission contains a wealth of information on the chemical and dynamical state of the Earth's atmosphere. However, observation strategies and data reduction processes are usually optimised to minimise the influence of these features on the astronomical spectrum. Here we discuss a measurement technique to optimise the extraction of the OH airglow signal itself from routine J-, H- and K-band, long-slit, astronomical spectroscopic observations. As an example, we use data recorded from a point source observation by the Nordic Optical Telescope's intermediate-resolution spectrograph, which has a spatial resolution of approximately 100 m at the airglow layer. Emission spectra from the OH vibrational manifold from v'=9 down to v'=3, with signal-to-noise ratios up to 280, have been extracted from 10.8 s integrations. Rotational temperatures representative of the background atmospheric temperature near 90 km, the mesosphere and lower thermosphere region, can be fitted to the OH rotational lines with an accuracy of around 0.7 K. Using this measurement and analysis technique, we derive a rotational temperature distribution with v' that agrees with atmospheric model conditions and the preponderance of previous work. We discuss the derived rotational temperatures from the different vibrational bands and highlight the potential for both the archived and future observations, which are at unprecedented spatial and temporal resolutions, to contribute toward the resolution of long-standing problems in atmospheric physics.

## 1 Introduction

### 1.1 OH airglow

The atmospheric region between 85 and 90 km represents a transition zone that lies between the thermosphere, where collisions are rare, and the collision-dominated, well-mixed mesosphere. This Mesosphere and Lower Thermosphere (MLT) region is

highly variable both chemically and dynamically (Smith, 2012; Smith, 2004). It is driven from above by diurnal, seasonal and long-term changes in solar insolation, and from below by tides, planetary waves and upward propagating gravity waves. Photochemical reactions in the MLT play a key role in the vertical distribution of energy. The major loss for mesopause ozone is its reduction to molecular oxygen via:

$$H+O_3 \rightarrow OH^* +O_2 \qquad\qquad (0.1)$$

This loss is balanced by the major source of the ozone, the combination of molecular and atomic oxygen via a mediator M:

$$O+O_2+M \rightarrow O_3+M, \qquad\qquad (0.2)$$

where the atomic oxygen is formed by the dissociation of molecular oxygen by solar radiation (Espy and Stegman, 2002; Sigernes et al., 2003). Reaction (0.1) is highly exothermic (>3eV), leading to production of vibrational levels from v'=7 to v'=9 of the OH product. Deactivation of these high vibrational states primarily occurs through photo-emission in the Meinel bands (Meinel, 1950), resulting in the bright OH airglow, localized in a thick (~8 km thick) layer near 90 km, that can be observed in the visible and infrared. Collisions of each long-lived vibrational excited state (v') with the surrounding gas effectively thermalize the lower, closely spaced, rotational states into a Boltzmann distribution (Pendleton et al., 1993). Hence, moderate resolution spectroscopic measurements of the relative population of the rotational levels of individual OH vibrational bands can be used to remotely sense the temperature of the mesopause region. Furthermore, the relative intensity of the individual vibrational bands can be used to estimate the relative populations of the v' states. Comprehensive reviews on ground based observations of OH and their applications to mesopause chemistry can be found in von Zahn et al. (1987), Yee et al. (1997), and Smith et al. (2010).

However, achieving high temporal resolution has often only been possible at low spatial resolution (and vice versa). In this paper we use J-, H- and K-band, long-slit observations of an astronomical point source made by the Nordic Optical Telescope near-infrared Camera and spectrograph (NOTCam). This intermediate-resolution spectrometer was used to obtain high quality, very high spatial (<100m) resolution observations with short integration times (~10s). We demonstrate how to extract and optimise OH atmospheric spectral data from these astronomical observations, and discuss the quality and validity of the derived data over the range of vibrational bands. Finally, we consider some problems of atmospheric physics that can be addressed with these new data.

## 1.2 Instrumentation

Founded in 1984, and located in La Palma (17°53' W, 28°45' N), the Nordic Optical Telescope (NOT) has a primary mirror with a diameter of 2560mm (Djupvik and Andersen, 2010). Its near-infrared camera and spectrograph, NOTCam (Abbott et al., 2000), has been used for imaging since June 2001 and spectroscopy since August 2003. The detector is a 1024×1024 pixel Rockwell Science Center HgCdTe "HAWAII" array. The low resolution mode ($R = \lambda/\Delta\lambda \sim 2100$) is sufficient to resolve the individual rotational lines of an OH vibrational level. The dispersing element is an echelle grism used together with broad band

filters to sort the orders. In low resolution spectroscopic mode the slit employed has a width of 128µm, corresponding to approximately 0.6 arcsec on the sky, and a slit length of 4 arcmin. At an approximate OH layer altitude of 87km (Baker and Stair Jr., 1988), 4 arcmin corresponds to around 100m on the sky. The detector has a dead-time of about 10 s due to reading out processes after each integration. Further details on NOTCam spectroscopy can be found in Telting (2004) and a description of the NOTCam in Djupvik (2001).

During point-source astronomical observations, measurements in the *J*-, *H*-, and *K*-bands (1.165-1.328 nm, 1.484-1.780 nm, and 2.038-2.363 nm, respectively) are typically taken. Although each of these wavelength bands is observed individually, atmospheric OH vibrational band transitions (7,4) and (8,5) are simultaneously observed in the *J*-band, the (3,1), (4,2), (5,3) and a part of the (6,4) in the *H*-band, and the (8,6) and (9,7) in the *K*-band (Meinel, 1950). The (9,7) band is of particular interest as it represents the highest vibrational level populated by reaction (0.1).

## 2 Data reduction

To demonstrate the data reduction procedures required to optimise the OH signal from routine astronomical observations, a single *H*-band spectroscopic exposure toward the star, 21 Vir (spectral type B9V, H = 5.64 mag), with an exposure time of 10.8s was used. This exposure time was chosen as an example as it is the shortest exposure time available in the archive, although the methods presented here are developed and optimized for a variety of conditions with integration times up to 600 s, which is the longest integration time available in the archive.

The H-band image presented here was taken in good astronomical observing conditions at an air mass of 1.516 at 05:50 UT on 19 February 2013, together with dark frames and flat fields recorded with a halogen lamp on the same night. The J and K band images presented in section 3.2 were recorded towards the same object with the same integration time within 5 minutes of the H band observation. The raw image frame is reproduced in Fig. 1(a). The vertical bright band in Fig. 1(a), located slightly to the left on the detector, is the *H*-band spectrum of the star. The weaker, nearly horizontal lines that curve upward are the atmospheric OH lines from the (3,1), (4,2), (5,3) and (part of the) (6,4) vibrational bands, listed from top to bottom in the image frame.

For the initial data reduction steps standard IRAF (Tody, 1993) astronomical procedures were followed. This entailed removing bad pixels (zero valued or cold) that are known and stable, and dark frames were used to create a pixel mask to remove hot pixels on the detector. The intensity of the OH lines is well within the linear range of the array, so non-linearity effects could be ignored. The dark current was removed using dark images taken toward the closed dome before observations began. Wavelength dependent variations in transmission and detector response were corrected by flat fielding using short integrations of a halogen lamp with a known colour temperature situated at the upper end of the optical path.

The stellar spectrum is much brighter than the OH lines and has to be removed. A linear mask, 60-pixels wide in the spatial direction and centred on the star's horizontal position, was applied to the detector frame. Sixty pixels removed the stellar

influence while maintaining as much OH data as possible. The remaining OH lines do not appear as straight lines on the array due to the intersection of the array detector with the telescope's diffraction optics. Instead they are parabolic, with a curvature that varies linearly with spectral position. To straighten them for integration, a line's spatial and spectral pixels (x,y') were mapped to a coordinate system where the line appears at the same spectral location, y, along the entire spatial extent, x, using the function $y = y'+(x-p_1)^2 \cdot p_2$. The detector is aligned such that the central position of the parabola, $p_1$, is constant at column number 465, and the curvature, $p_2$, is given by $p_2 = 3 \cdot 10^{-5} + 5.3 \cdot 10^{-8} \cdot y'$. The result after processing is shown in Fig. 1(b).

Given this transformation, the OH lines could then be integrated in the spatial dimension, x, for a given value of wavelength, y, with the standard deviation used to estimate each line's uncertainty, to form a high signal-to-noise spectrum (after accounting for the pixels masked out as described above). Gaussian functions with a full width half maximum of 0.42, 0.73 and 0.87 nm were found to fit the line shape of the resulting OH lines in the centres of the *J*-, *H*- and *K*-bands, respectively, to within the noise present in the data. The fitting however is performed in units of pixels rather than nm. The line width is constant in pixel units for each band separately at 2.27 pixels.

Since the wavelengths of the OH lines are known (Rothman et al., 2013), and the lines are easy to identify, the abundant lines themselves were used for wavelength calibration. The brightest lines of the Q-branch and the six brightest P-branch lines of each transition were used for the calibration of each filter band. For the *H*-band example presented here, this led to a calibration based on 22 pixel/wavelength pairs (only the main Q-branch lines were used for the (6,4) transition). The wavelength calibration has only small nonlinear contributions.

## 3 Results

### 3.1 *H*-band data

In Fig. 2 the resulting *H*-band spectrum is shown based on the total integration time of 10.8s. The spectrum has been corrected using a relative spectral radiance calibration derived from the flat-field source. However, an absolute calibration could be achieved using observations of a standard star of known intensity. The absolute calibration is however not needed for this work as we are only interested in the hydroxyl temperatures. The flat fielding distorts the edges of the spectrum slightly as the filter transmission nears zero in the red highlighted regions in the figure. At the long-wavelength end, the Q- and R-branches from the (6,4) transition can be seen. The other three resolved bands belong to the (3,1), (4,2) and (5,3) transitions. The signal-to-noise ratio is 280 for the Q-branch lines and around 200 for the P-branch lines for this 10.8 s exposure. Longer integration times increase these ratios further.

As can be seen in Fig. 2, the background is not completely flat but has low frequency variations that span wavelength ranges greater than the OH line widths. Since these background variations will influence the relative intensities of the individual rotational lines, they impact the fitted OH rotational temperature since it is strongly dependent on the relative heights of the

lines. For this reason a 7-order Butterworth IIR filter with a cutoff (-3dB point) at $\frac{1}{3.3nm}$ was applied to the extracted 2D spectra to remove the low frequency variability. This filter was optimised through repeated tests on synthetic OH spectra to have minimal impact on the temperatures derived from the OH rotational lines. Fig. 3a and b show the (5,3) vibrational band before and after application of this filter. After filtering, the spectrum was normalized such that the sum over the absolute values of the spectrum was equal to one.

A model spectrum was created using the OH line strengths, $S_{j'j''}$ for a given vibrational transition, tabulated in the HIgh-resolution TRANsmission (HITRAN) molecular absorption database (Rothman et al., 2013). The number of photons emitted in each rotational transition, assuming a Boltzmann distribution of population in the upper-state rotational levels, is given by:

$$I = N_{v'} \cdot S_{j'j''} \cdot \left( \nu_{j'j''} \right)^3 \cdot \exp \left( - \frac{E_{j'}}{k T_{\text{rot}}} \right)$$

Here $N_{v'}$ is the relative population of the upper vibrational level, v', and $\nu_{j'j''}$ is the wavenumber of the transition from the upper state level at potential energy $E_{j'}$ to the final rotational state at energy $E_{j''}$. To form the model spectrum, these relative line intensities were convolved with the instrumental line shape, then filtered and normalized in the same manner as the data. A $\chi^2$ minimisation using the rotational temperature and a total intensity scaling factor as the variable parameters was then performed between the model and the data to determine the rotational temperature best fitting the data. A Brent algorithm (Brent, 1973) from the GNU Scientific Library (Galassi and Gough, 2009) was used for the minimisation. Following Pendleton et al. (1989), only rotational lines originating from levels $N \leq 4$ were used in the fitting, since higher levels may not be thermalized. The temperature fitting routine was tested against model data with added Gaussian white noise where temperatures were varied between 150 and 400 K. The resulting fit was found to reproduce the input rotational temperatures to within the fitting errors of the temperature parameter.

Fig. 3(c) shows the data from Fig. 3(b) along with the fitted model spectrum shown as the red line. The fitted temperature in this case is found to be 186.5 ± 0.7 K, representing a 0.4% relative uncertainty. The residual spectrum (data - fit) is shown in Fig. 3(d), where it may be seen that although small differences in the wavelength calibration or line shape may occur, the fitted spectrum accurately represents the observation. This and similar tests for the other Meinel bands and different integration times demonstrate that the model used is robust, that fitting the high pass filtered data works well, and that the fit to the data converges to a temperature with a small error.

## 3.2 Other vibrational bands

Fig. 4 shows an overview of all the measured OH vibrational transitions recorded in the *H*-, *J*- and *K*-band spectra taken toward the same star, extracted in the manner outlined above. The measurements in the *J*- and *K*-band were executed at 05:53 UT and 05:47 UT respectively, meaning that the total time between all observations was about 6 minutes. All spectra were taken with the same 10.8s integration time and are normalised to the same Q branch intensity for comparison. For the majority of

vibrational transitions, the signal is very much greater than the noise, providing excellent data for temperature fitting, especially for the transitions in the H and *K* bands.

Since the (7,4) and (8,5) transitions in the *J* band have a $\Delta v=3$, the lines are not as intense as in the other $\Delta v=2$ bands. Due to the lower intensities of these $\Delta v=3$ bands, the signal to noise is substantially reduced for this short integration which is typical of the NOTCam data. As a result, the fitting error for these bands was very large. While longer exposures, or the co-addition of sequential short exposures, would improve the fitting error of the OH (7,4) band, the R and Q branches of the (8,5) band are overlapped by the optically thick $O_2$ Infrared Atmospheric band at 1270 nm, and the filter cut-off reduces its $P_1(4)$ line to near the noise level. Thus, the temperatures for the (8,5) bands would remain compromised even for longer integrations.

### 3.3 Temperature gradient

The peak concentrations of the neighbouring vibrational levels are, on average, separated in altitude by 0.5 km (von Savigny et al., 2012). Even though the absolute peak altitudes are known to vary with season (Gao et al., 2010), they can be taken as constant on the time scales of a few minutes over which this experiment was executed. With a steady-state OH model driven using a neutral atmosphere from the Naval Research Laboratory Mass Spectrometer and Incoherent Scatter Radar Empirical model (NRLMSISE) (Picone et al., 2002), the altitude of the (9,7) transition was fixed. The relative altitudes of the other vibrational transitions were then assigned using an altitude separation of 0.5 km, the average separation found by von Savigny et al. (2012). The rotational temperatures derived from each of the individual vibrational bands then provide an estimate of temperature gradients present similar to what has been done by Perminov et al. (2007) and Schubert et al. (1990).

Fig. 5 shows the distribution in altitude for the data presented here together with the NRLMSISE model kinetic temperature for the corresponding location and time. In addition, the NRLMSISE profile has been smoothed with the volume emission rate profile of an OH band derived from the steady-state OH model to reflect the effect of the OH layer width on the rotational temperatures. The temperature profile from this model is consistent with the gradients estimated from our data for the (3,1), (4,2), (5,3), (6,4) and (9,7) transitions. It is important to note that the *J*-, *H*- and *K*-band data shown in Fig. 5 were acquired using sequential 10.8 s observations, spanning only 6 minutes. Thus, large deviations from the climatological gradients can occur due to wave activity (Xu et al., 2000). While the agreement here may be fortuitous, a time sequence of short integrations may give insight into the wave-induced temperature gradients.

While the (8,6) transition is anomalously high, it was found that the P(2) and P(4) lines are partially absorbed by atmospheric $H_2O$ and $CO_2$ (Jones et al., 2013; Noll et al., 2012). To model the impact of this atmospheric absorption, synthetic spectra with rotational temperatures between 130 K and 300 K were created. When these were fitted with the technique described above, this input temperature could be retrieved. When these synthetic spectra were however first multiplied by a high resolution (0.002nm) absorption spectrum for seasonally averaged conditions obtained from the Cerro Paranal sky model (Jones et al., 2013; Noll et al., 2012), the fitted temperature is approximately 8% higher than the original synthesized temperature. This

would account for the higher fitted temperature of the observed (8,6) band shown in Fig. 5. Using this same technique, the temperatures for all other vibrational-rotational transitions presented in Fig. 5 were examined and found not to be significantly affected by atmospheric absorption.

With the intent of demonstrating that the NOTCam provides atmospheric data that can be used to supplement other astronomical data sets used for aeronomic studies (e.g Osterbrock et al. (1996), Noll et al. (2015), Cosby and Slanger (2007)), a sample, short-integration spectrum has been analysed, and both the analysis procedure and results presented. With the exception of the weak or compromised v'=7 and 8 levels discussed earlier, the temperature variation with vibrational level observed using the NOTCam and analysed here reflects the MSIS kinetic temperature to within two sigma. This is consistent with previous observations using ground-based spectrometers or interferometers (e.g.Innis et al. (2001); Oberheide et al. (2006); French and Mulligan (2010); and Dyrland et al. (2010)). However, it must be pointed out that the NOTCam measurements cover a very short time span and may be affected by gravity-wave perturbations of the climatological temperature gradient of the atmosphere represented by MSIS. The temperature distribution with vibrational level observed here shows a small decrease toward higher levels, although this decrease is, apart from the v'=9, not significant at the two sigma level. This is at odds with the measurements of Noll et al. (2015) and Cosby and Slanger (2007), who show strong increases in temperature with vibrational level that they attribute to non-thermodynamic equilibrium effects. However, our results are consistent with the results of Lübken et al. (1990), Espy and Hammond (1995), Wrasse et al. (2004), and Perminov et al. (2007), who show similar or decreasing temperatures with increasing vibrational level. Once again, the short duration of this demonstration data is not able to resolve this apparent discrepancy, but the larger NOTCam data set may prove useful in this regard.

**4 Outlook**

Near-IR spectroscopy with NOTCam has been performed over the last 13 years toward point-like sources (stars or marginally extended objects). NOTCam, in both spectroscopic and imaging mode, has been mounted and used on the telescope an average of 15% of the observing time between 2003 and 2016, spread evenly throughout the calendar year. Thus, good seasonal coverage is available as indicated in Fig. 6, which shows the distribution of dates between November 2007 and June 2016 when the NOTCam was used for long-slit spectroscopic observations.

The spectroscopic mode has been continuous and stable with no change of optical elements, except for the addition of the two broad-band filters, Z and Y, in 2010. There has been a change of detector, most recently in 2007, but each array is well characterized. All data older than one year (the proprietary period at the NOT) are available in the NOT data archive. Fig. 7 shows histograms of the total number of spectroscopic observations taken in the *J*-, *H*- and *K*-bands for each hour indicating nearly uniform coverage in each wavelength region.

With this data set, a variety of atmospheric problems can be addressed. One example is to simply generate climatological temperature gradients of the mesopause region from the long-term data as in Fig. 5. High quality, long-term observations of the mean state, trends and inter-annual variability of this region are rare, especially at low latitudes, and can serve as an important standard against which whole atmosphere models can be validated.

Additionally, there is a long-standing discussion as to whether the vibrationally excited OH quenches to the ground state ("sudden death"), or relaxes to the next-lowest vibrational level (McDade, 1991). Knowledge of the population and quenching of the individual upper states is essential for the interpretation of the OH airglow emission for remote sensing of the mesopause region. These high quality data described above, together with a steady-state model, can be used to estimate the ratio of single to multi-quantum quenching efficiency accurately. Although these quenching rates have been examined by Xu et al. (2012)

using broad-band SABER measurements that combine multiple high-v' and low v' levels, the individual bands resolved by astronomical telescopes would allow an unambiguous assessment of these rates.

The data discussed here have very short integration times as a demonstration of the worst-case conditions. This means that a series of measurements of the same spot of the sky can scan the OH layer with a repetition rate of down to 20 seconds. This lies well below the upper boundary of gravity waves (Kovalam et al., 2011) and stretches into the domain of acoustic and

15 infrasonic waves. Although these waves have been observed in the hydroxyl airglow (Bittner et al., 2010; Pilger et al., 2013), the low spatial resolution of the observations creates ambiguity in their identification. Here the observations have a spatial resolution on the airglow layer of about 100m (which is 4 arcmin), sampled with approximately 1000 pixels in the spatial direction of the NOT detector. This resolution is achieved with the telescope in "staring mode", which means without star tracking. The resolution with star tracking is dependent on the slit orientation and is in worst case up to 6.7 x 2.7 arcmin over

20 a 10.8 s integration. Using staring-mode, the high resolution in both the temporal and spatial dimensions allows high frequency waves to be measured at a commensurate spatial resolution. This opens up new possibilities to study the smallest structures of waves propagating through the OH layer in the MLT. While staring observations would recover the full spatial resolution of the instrument, care should be taken with the slit orientation and choice of direction to ensure that bright astronomical objects do not interfere with the airglow observations.

Finally, in parallel with the atmospheric work, it will be possible to quantify how the intensity of the mesospheric OH background in the astronomical *H-*, *J-* and *K-*bands varies on different timescales (from minutes to years) over La Palma. This will be useful in the planning and scheduling of observations in order to optimize dithering strategies and observing modes, especially important for instruments observing simultaneously at optical and infrared wavelengths.

**Acknowledgements**

This work was based on observations made with the Nordic Optical Telescope, operated by the Nordic Optical Telescope Scientific Association at the Observatorio del Roque de los Muchachos, La Palma, Spain of the Instituto de Astrofisica de

Canarias. We thank the staff from the NOT for their help and support during a visit by CF in November 2014. This work was supported by the Research Council of Norway/CoE under contract 223252/F50. Archive data from the NOTCam can be accessed via http://www.not.iac.es/observing/forms/fitsarchive/.

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

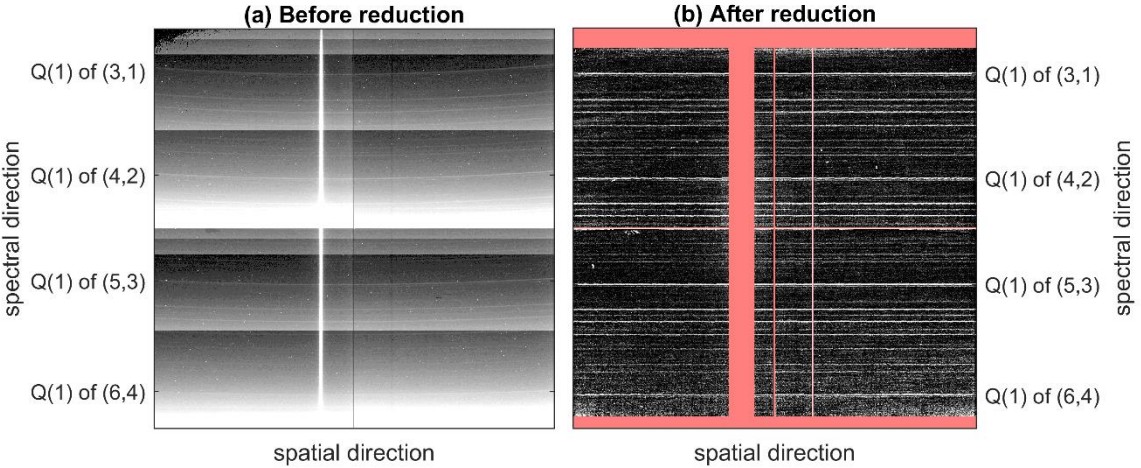

Figure 1: (a) The array in its raw form. The bright star is visible as a vertical band slightly to the left of centre. The curved OH lines are visible in the background. (b) The array after cleaning and flat fielding, with the star removed and the OH lines straightened. The unusable parts of the array are blanked out in red, and the OH lines can be seen clearly.

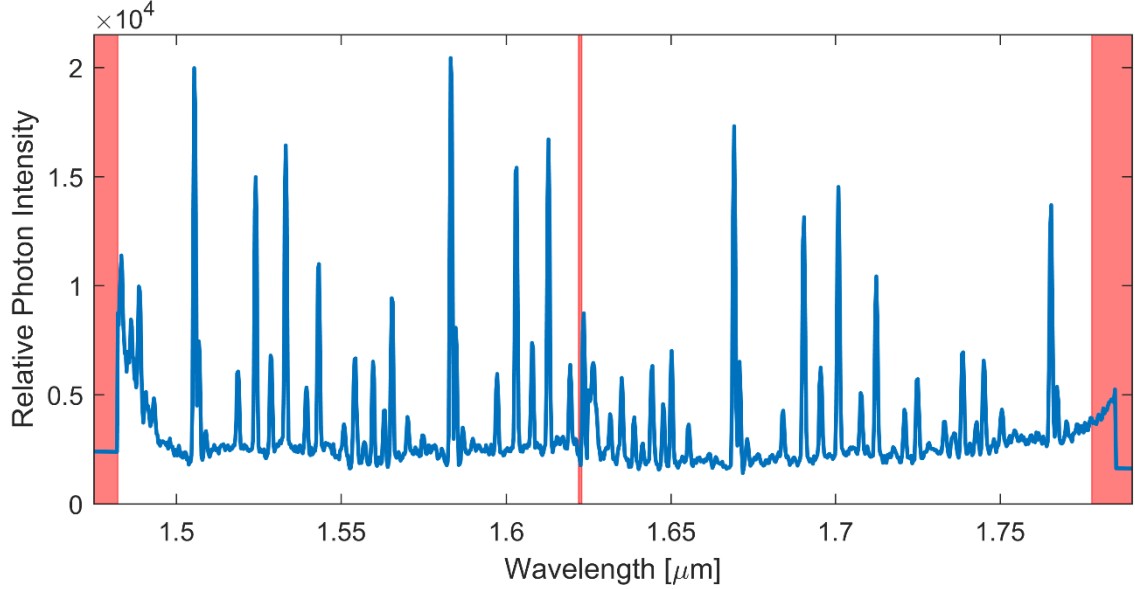

Figure 2: *H*-band spectrum of atmospheric OH extracted from the data frame presented in Fig. 1. Red shading highlights parts of the spectrum not used in the subsequent rotational temperature fitting.

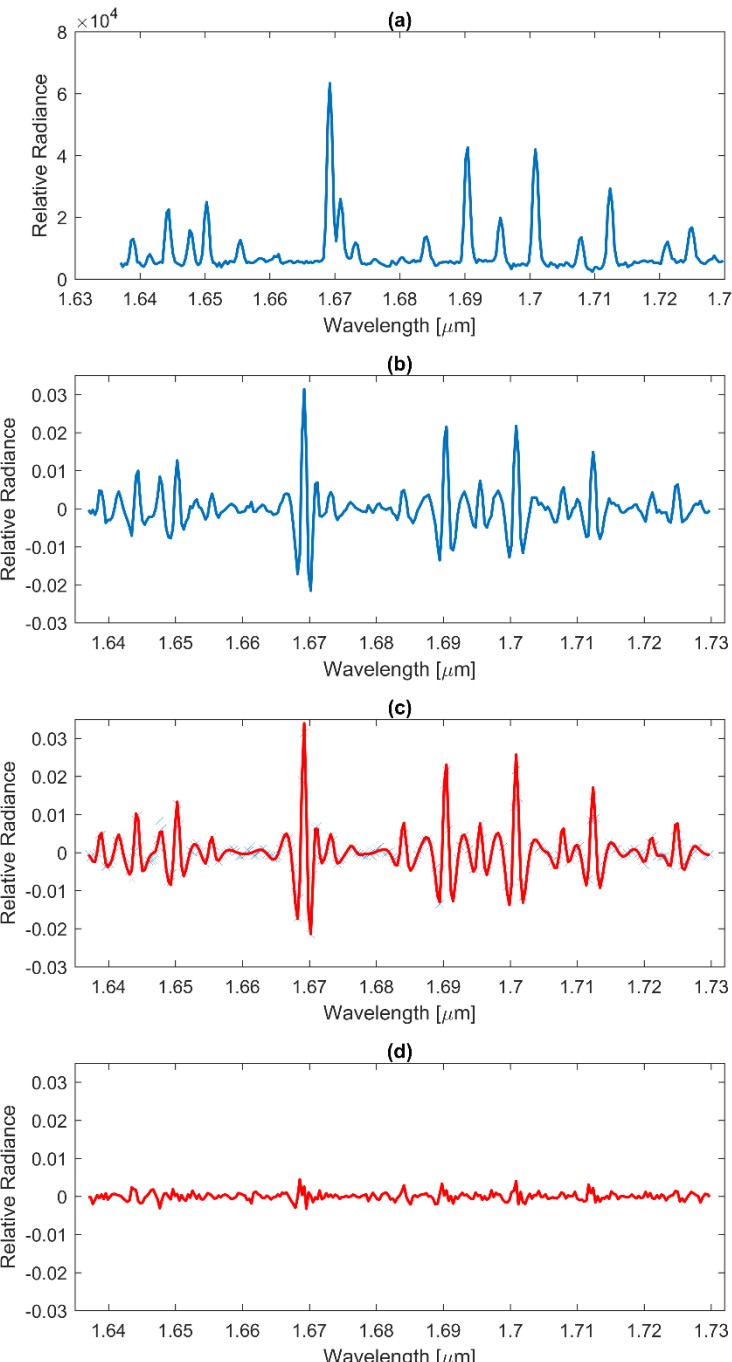

**Figure 3: (a) Unfiltered spectrum of the (5,3) transition. (b) Filtered spectrum of the (5,3) transition. (c) Normalised spectrum from Fig. 3(b) (blue crosses) with the fitted, filtered model-spectrum shown in red. The fitted rotational temperature is 186.5 ± 0.7 K. (d) Residuals of the fit from Fig. 3(c).**

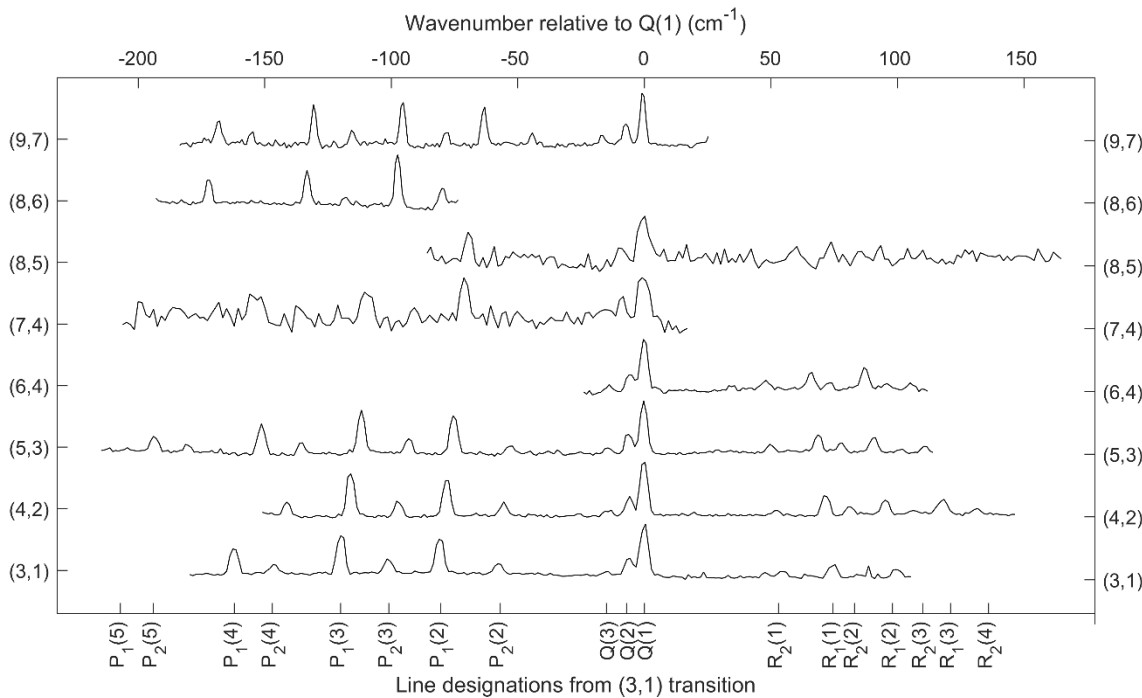

**Figure 4: Spectra from the *H*-, *J*- and *K*-bands in wavenumbers relative to the Q(1) line in each band. The Q(1) lines of each spectrum are aligned. Line positions for the (3,1) transition are shown..**

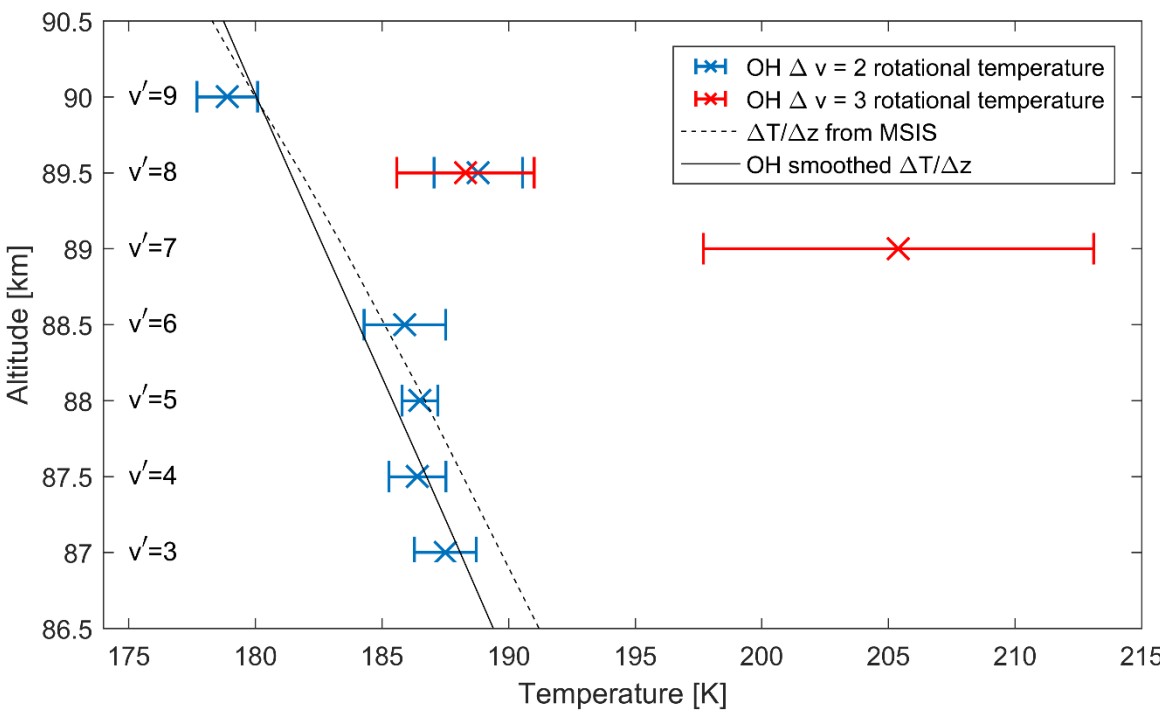

**Figure 5: The temperatures from the data presented in Fig. 4 with Δv=2 in blue and Δv=3 in red. Both are presented with their one sigma errorbar. The altitude of the (9,7) transition was fixed with a steady-state model using a neutral atmosphere. The altitudes of the other transitions are fixed with a relative distance between neighbouring transitions of 0.5 km, following von Savigny et al. (2012). The NLRMSISE temperature gradient for 2013 February 19 at La Palma (17°53'W, 28°45'N) at 05:50 UT is shown as a black, dashed line. The same gradient but smoothed with a modelled OH distribution in altitude is shown as a black solid line.**

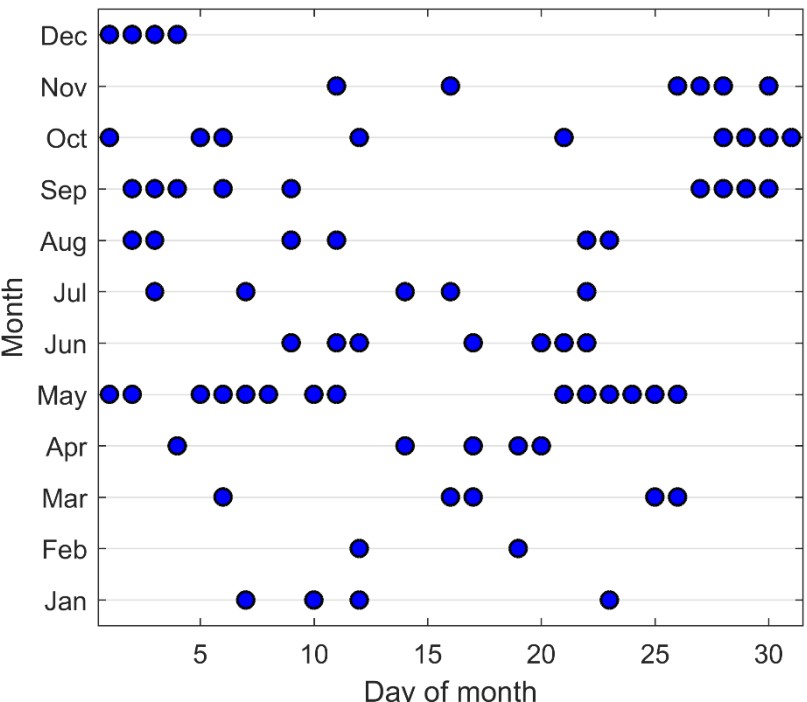

**Figure 6: Dates between November 2007 and June 2016 (which is the coverage in the NOTCam archive) when the NOTCam has been mounted at the NOT and making spectroscopic astronomical observations.**

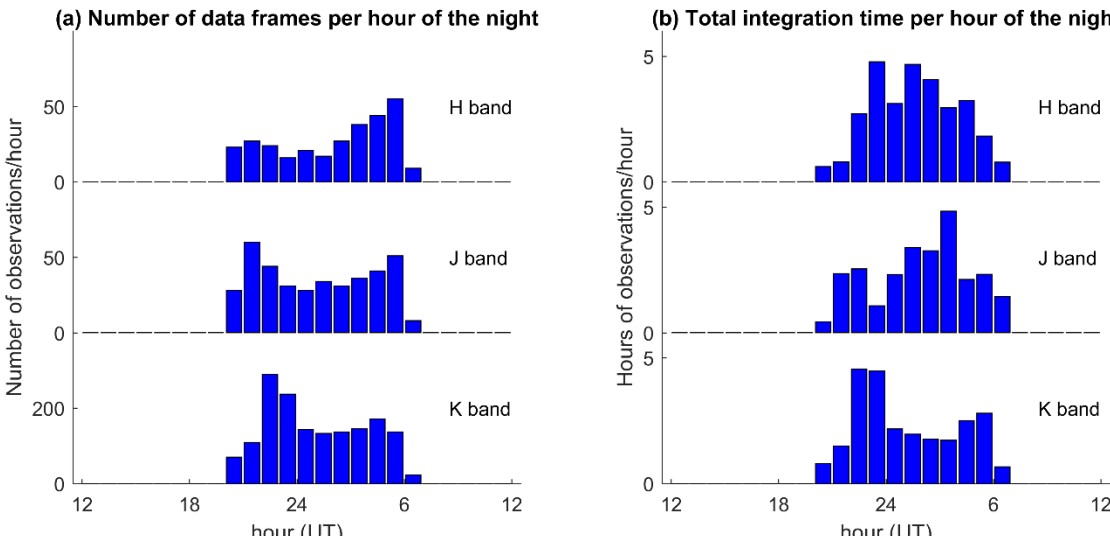

**Figure 7: (a) Total number of spectroscopic data frames recorded in each hour and each band with NOTCam between November 2007 and June 2016.(b) Total hours of on-target integration time for spectroscopic data recorded in each hour and each band with the NOTCam in the same period of time.**

