# Peer review of "Optimising hydroxyl airglow retrievals from long-slit astronomical spectroscopic observations"

_Atmospheric Measurement Techniques, 2017_

## Referee Comment (RC1) · Anonymous Referee #2 · 5 Apr 2017

Review of the manuscript "Optimising hydroxyl airglow retrievals from long-slit astronomical spectroscopic observations" by Franzen et al.

Comments: The terrestrial atmosphere airglow is usually regarded as contamination of astronomical spectroscopy observations. However, these high resolution airglow spectroscopy observations are very useful for middle and upper atmosphere researches. In this work the authors extracted OH airglow spectra from the background astronomical observations and gave the process of the rotational temperature derivation from the extracted spectra. Moreover, the rotational temperatures of different vibrational bands characterized different heights are discussed. The observations at high spatial and temporal resolutions could contribute to resolve some longstanding problems in atmospheric physics. This paper is well written and is suitable published. I suggest that this high quality data can be used to study the long term variation of the mesopause region, such as, response to the solar cycle, in the future.

---

## Referee Comment (RC2) · Anonymous Referee #1 · 11 Apr 2017

General comments:

The paper describes a feasibility study for the use of astronomical long-slit spectra from the NOTCam instrument of the Nordic Optical Telescope (La Palma, Spain) for OH airglow investigations. The data properties, data reduction, and analysis in terms of the derivation and interpretation of OH rotational temperatures are described based on a set of three 10.8 s exposures covering the wavelength ranges related to the photometric J, H, and K bands. Furthermore, the paper discusses the available NOTCam data sets and the OH-related scientific questions which can be investigated using such spectra.

The study is interesting since it gives an overview of the challenges and prospects

related to the use of long-slit astronomical spectra for atmospheric research in the mesopause region. However, the paper is fairly short, which makes it difficult to fully understand the discussion. In particular, it is not clear whether the systematics related to the OH rotational temperature measurements are treated in a proper way. In any case, the study neglects the influence of non-LTE effects on the derived temperatures. The following list of specific comments is relatively long for such a short paper. This means that the manuscript has to be significantly modified to allow its final publication in AMT.

Specific comments:

P.1, L.16 and P.2, L.21: It would be better to use the term "spectrograph" instead of "spectrometer" in order to clarify that all covered wavelengths are recorded simultaneously.

P.3, L.3: It is not clear what "standard astronomical observations" means. Is the word "standard" related to "standard star" on L. 9 or does it refer to a certain observing mode? Are there also non-standard observations? Does this classification affect the size of the suitable data sample?

P.3, L.3: It would be good to explicitly state that the three bands are observed independently with a certain time delay. NOTCam offers more than the three mentioned filters, which could be combined in various ways. Why do the authors only list the combination of J-, H-, and K-band observations? Finally, I recommend to describe the wavelength ranges covered by the different modes. There will be readers who are not experienced with astronomical filter bands.

P.3, L.8: I can understand that only a single spectrum is used for the illustration of the data reduction procedure. However, the discussion in almost the entire paper (except for the outlook) is based on a single series of three 10.8 s exposures (each in a different band). If the whole data reduction and analysis is optimised for these spectra, it is not clear how changes in the observing mode (filter band, spectral resolution, expo-

sure time), observing conditions (atmospheric transmission and turbulence), and the calibration data (with a possible impact on the robustness of the reduction) could affect the quality of e.g. the OH rotational temperatures.

P.3, L.9: The given star is probably a telluric standard star, i.e. the corresponding spectra are used to correct atmospheric molecular absorption. Is there a special reason for this selection? In principle, the observations of astronomical science targets or spectrophotometric standard stars could have been used as well.

P.3, L.10: Why is the exposure time of the example only 10.8 s? This causes uncertainties in the line measurements and hence the OH rotational temperatures, which could be avoided by longer exposures. Of course, large time differences can reduce the coupling between the OH intensities in the spectra of different filter bands which are taken consecutively. Nevertheless, observing times distinctly longer than 10.8 s would still be well below the Brunt-Väisälä period. Was the selection by purpose (considering the arguments given above) or were there restrictions with respect to the data that was provided by the NOT for this study?

P.3, L.31: "A Gaussian function with a width of 0.31 nm": What was used for the J- and K-band observations? For constant resolving power, the line width is proportional to the wavelength. Moreover, the resolving power could/should depend on the observing mode.

P.4, L.8 and Fig. 2: The reduced spectrum does not seem to be flux calibrated. Fig. 2 shows ADU as intensity units. It can lead to a significant bias if such a spectrum is used for the derivation of OH rotational temperatures. Flat-fielding is only for the correction of small-scale variations. For the slowly varying intensity variations and the absolute calibration, instrument response curves derived from observations and reference tables of spectrophotometric standard stars are required. Only the effect of atmospheric extinction by scattering can usually be neglected in the near-IR.

P.4, L.11: How were the signal-to-noise ratios estimated? Poisson noise?

[Figure]

P.4, L.17: The continuum underlying the OH emission is removed by applying a Butter-worth filter. This is not the most obvious way to separate line from continuum emission. Were therefore also other approaches tested? For example, continuum windows can be defined on both sides of the interesting lines. Afterwards, the continuum at the line positions can be interpolated and subtracted.

P.4, L.19 and Fig. 3: It is stated that the Butterworth filter should have a "minimal impact" on the OH lines. However, Fig. 3 reveals that there are major changes in the line shape. Strong negative residuals occur. Do this complex features reflect the true line intensities in a reliable way? This could be tested using alternative approaches (see comment on P.4, L.17).

P.4, L.20: The resulting spectrum is normalised to 1. However, it varies around 0. Hence, is this approach sufficiently robust?

P.4, L.22: Fitting a model spectrum instead of direct line measurements can cause relatively large systematic errors if the model assumptions in terms of airglow physics and the adaption to the observed data (e.g. line-spread function) are not sufficiently accurate. Was this approach chosen because of the relatively low spectral resolution?

P.4, L.22: "known OH line strengths": The Einstein-A coefficients for OH lines are not that well known. Ratios of Einstein-A coefficients for lines from different OH bands can easily vary by a factor of 2 or more depending on the set used. This study appears to be based on those from the HITRAN database, i.e. Goldman et al. (1998). Have the authors ever considered to also use OH molecular parameters from other publications, e.g. van der Loo & Groenenboom (2007, 2008) or Brooke et al. (2016) (to list relatively recent papers). Since this study combines P-, Q-, and R-branch data, the choice of the coefficients can be critical for comparisons of rotational temperatures from OH bands with different line sets (more critical than in the case of P-branch lines only).

P.4, L.23: The assumption of a Boltzmann distribution with a single temperature is a strong simplification. In fact, any change of the line set will affect the resulting temperature (see Noll et al. 2015). Apart from uncertainties in the Einstein-A coefficients, this is caused by line-specific amounts of non-LTE contributions. They vary depending on vibrational level, rotational level, electronic substate, and observing time. Any OH rotational temperature is a pseudo-temperature (deviating from the kinetic temperature). Whether this is critical for a scientific study is a different question. In any case, one should take OH rotational temperatures and related comparisons with care.

P.4, L.28: The success of the model fit also depends on the accuracy of the instrumental line shape. Are the model spectra calculated using the simple Gaussian with fixed width mentioned at L. 31 of P. 3? Have the authors checked how the results on the OH rotational temperatures depend on the line-spread function (within the estimated uncertainties)?

P.5, L.1: "higher levels may not be thermalized": This is also likely for lower levels (see comment on P.4, L.23).

P.5, L.3: Is "accurate" the correct word to describe the quality of the OH rotational temperatures? As discussed before, the systematic errors related to the resulting temperatures can be relatively large. Uncertainties in the Einstein-A coefficients, non-LTE effects, line-spread function, and also atmospheric absorption (see comment on P.5, L.32) can be critical.

P.5, L.5: An error of 0.56 K is remarkably small. How was it calculated?

P.5, L.6: "may not be completely Gaussian": This is not unexpected. For example, the Gaussian might be convolved with a boxcar, which considers the influence of the entrance slit.

P.5, L.7: "the model used is robust": Note that this statement applies to the spectrum that was used to optimise the analysis. For a different observation, this might not be true anymore (even if the unconsidered systematic model errors are neglected).

P.5, L.11: Fig. 4 shows relatively large errors of the wavelength calibration. In part, the

shifts are higher than a line width. In particular, the spectrum of OH(7-4) does not fit well. Another example is the mismatch of the highest P-branch lines of OH(8-6) and OH(9-7). What is the influence of these wavelength calibration errors on the model fit and the corresponding rotational temperatures?

P.5, L.17: "longer integrations ... would enhance the data quality": The authors assume that the 10.8 s exposure in the J band is too short for reliable temperature measurements. Why was this not tested using a sufficiently long exposure? Only in this way, it is possible to distinguish statistical from systematic errors.

P.5, L.20: Assuming a fixed separation for layers of adjacent vibrational levels of 0.5 km can only be a very rough estimate since this difference is highly variable and also depends on which vibrational levels are compared (e.g., von Savigny et al. 2012; Xu et al. 2012). Hence, Fig. 4 essentially shows OH rotational temperatures as a function of vibrational level, even if the ordinate provides altitudes in kilometres. It would be good to clarify that the resulting temperature gradients are only qualitatively correct.

P.5, L.25: "the altitude [sic] ... were assigned": The fixed altitude step size of 0.5 km could be mentioned here again.

P.5, L.26: The term "atmospheric temperature profile" can be misleading. Apart from the already mentioned uncertainty in the true emission peak altitude (comment on P.5, L.20), it also has to be considered that this profile is strongly smoothed due to the typical emission layer widths of 8 to 9 km. The most critical issue is the fact that the given rotational temperatures are a combination of kinetic temperatures and non-LTE effects. However, the term "atmospheric temperature profile" suggests that a kinetic temperature profile is shown.

P.5, L.27: For a comparison of the NOTCam data and the NRLMSISE model, the latter should be smoothed considering a typical OH emission profile. This should significantly reduce the model temperature gradient (see Noll et al. 2016).

P.5, L.28: In view of all the effects which were not considered for the temperature comparison, I do not think that a safe statement on possible similarities can be made. In principle, there should not be an agreement between the observed and the modelled data due to the contributions of non-LTE effects to the former.

P.5, L.29: It would be better to show the temperatures from the OH(7-4) and OH(8-5) bands. Otherwise one could think that the authors want to hide something. Even if the quality of the rotational temperature measurements for these two bands are lower, they can be plotted if realistic error bars are assumed. There would not be an issue with the signal-to-noise ratio of these observations if longer exposures were taken for this paper (see also comment on P.5, L.17).

P.5, L.30: In view of all the effects which were not considered for the temperature comparison, it cannot be stated that the temperature for OH(8-6) is "anomalously high". Moreover, it is even expected that the values for the 8th vibrational level are the highest (Cosby & Slanger 2007; Noll et al. 2015, 2016).

P.5, L.31: "P(2) and P(4)": The numbers for the electronic substates are not indicated here. What was exactly done to identify that these lines are partially absorbed? The given references suggest that the Cerro Paranal sky model could have been used for this.

P.5, L.32: How were the correction factors for the line intensity decrease in the model estimated? In principle, one needs a very high resolution atmospheric transmission spectrum (resolving power $> 10^6$) for realistic atmospheric conditions. In particular, the amount of water vapour is highly variable. Therefore, it is important to estimate this amount if selected lines are affected. Apart from data from global weather models, direct measurements using the observed data are most promising. The telluric absorption in the observed stellar spectrum can be investigated with suitable fitting tools. Fortunately, the P-branch lines in the OH(8-6) band are mostly affected $CO_2$, which is much less variable than water vapour. Nevertheless, the high resolution transmission spectrum and a model of the airglow emission line profiles are needed to derive the correction factors. There are several papers on this topic (e.g., Espy & Hammond 1995; Noll et al. 2015; Chadney et al. 2017). Patrick Espy is even a co-author of this paper.

P.6, L.1: A change of the temperature by 8% is a huge effect, which can cause large uncertainties depending on the quality of the line absorption derivation. More details are needed to evaluate the uncertainties in the resulting OH rotational temperatures.

P.6, L.2: It is not clear what "same technique" means since the procedure for the line absorption correction is not described (see also comment on P.5, L.32).

P.6, L.4: This statement could change (see also comment on P.5, L.28).

P.6, L.5: The authors provide a long list of papers which apparently support their conclusion of decreasing OH rotational temperatures with increasing vibrational level. However, Oberheide et al. (2006), French & Mulligan (2010), and Dyrland et al. (2010) only provide results for a single OH band, which cannot be used for this investigation. In the case of Lübken et al. (1990), the results for OH(8-3) and OH(3-1) are quite similar. However, they are based on two independent instruments with different measurement periods. Sivjee & Hamwey (1987), Phillips et al. (2004), and Wrasse et al. (2004) compare rotational temperatures from the OH(8-3) and OH(6-2) bands. The first two studies find higher temperatures for OH(8-3), which is in agreement with the uncorrected NOTCam results but disagrees with the authors' conclusions. Espy & Hammond (1995) find similiar temperatures for OH(7-4) and OH(3-1), which also does not support the conclusions. Only the northern winter measurements used by Perminov et al. (2007) appear to be in good agreement.

P.6, L.8: The list of publications finding an increase of the rotational temperatures with vibrational level could be complemented by Cosby & Slanger (2007). This investigation was the first one which clearly showed this remarkable temperature dependence with a maximum for the 8th vibrational level, which can only be explained by a significant

contribution of non-thermalised rotational populations.

P.6, L.12: NOTCam competes with other instruments at the NOT. Nevertheless, an observing time fraction of 15% is relatively small. Moreover, it has to be considered that NOTCam is also an imaging instrument and that there are various spectroscopic modes with different wavelength coverages. Therefore, the number of spectra that cover a given OH band should be relatively small. This means that it is unlikely that a good annual coverage is achieved by the scheduled observations. It appears to be necessary to combine data from the whole period between 2007 and 2016 to perform variability studies. Exceptions could be individual observing nights with a good time coverage. However, this depends on the design of the corresponding astronomical project if archival data are used. Would it possible to perform dedicated airglow observations? I would like to see a more detailed discussion of the data set in terms of such limitations.

P.6, L.13 and Fig.6: The figure lacks information on the total number of spectra for a certain day of year and in which years these observations were performed. One could use more symbols, symbol sizes, and/or colours to provide more details. The figure only shows data taken between November 2007 and June 2016. Were there no suitable observations before this period although the spectroscopy mode started in August 2003 according to P. 2, L. 29?

P.6, L.18: Apart from showing the histograms in Fig. 7, it would be helpful if the total numbers of spectra for the different bands are provided in the text. Since the paper discusses a set of consecutive observations in three bands, it would also be interesting to know how many combined observations of this kind exist in the data archive. As the exposure time seems to be an issue at least in the J band, it would be good to learn more about the distribution of exposure times.

P.6, L.20: I am not sure whether this rather limited data set can be called "extensive".

P.6, L.31: Does the NOTCam archive already include suitable time series with high
temporal resolution or would it be necessary to apply for observing time for a dedicated study?

P.7, L.3: In the case of observations of astronomical objects, the spatial resolution is effectively reduced by the moving telescope. This issue should be discussed. Would it be possible to perform NOTCam observations without telescope tracking? However, if this was possible, there would be possible issues related to the data reduction of 2D spectra with various star traces.

Technical corrections:

P.5, L.25: altitude -> altitudes (see comment on P.5, L.25)

P.6, L.5: It would be better to write "northern winter" (or something similar) instead of only "winter".

P.6, L.8: measurement -> measurements

P.14, Fig.5: In the coordinates for La Palma, the "N" for northern is missing.

P.7, L.2: "large spatial resolution" -> "low spatial resolution"

---

## Author Comment (AC2) · 22 Jun 2017

We thank the reviewer for the thorough feedback. We will answer each point in the following.

*P.1, L.16 and P.2, L.21: It would be better to use the term "spectrograph" instead of "spectrometer" in order to clarify that all covered wavelengths are recorded simultaneously.*

This is true and has been corrected in the paper.

*P.3, L.3: It is not clear what "standard astronomical observations" means. Is the word "standard" related to "standard star" on L. 9 or does it refer to a certain observing*

[Figure]

*mode? Are there also non-standard observations? Does this classification affect the size of the suitable data sample?*

In this case, "standard" was used to mean "typical". In the processing, the star intensity in the spatial direction is removed out to the level where the stellar contribution is indistinguishable from the background (to within the noise level), the actual type of star is inconsequential for the observation. To avoid confusion, "standard astronomical observations" has been replaced by "typical, point-source astronomical observations".

*P.3, L.3: It would be good to explicitly state that the three bands are observed independently with a certain time delay. NOTCam offers more than the three mentioned filters, which could be combined in various ways. Why do the authors only list the combination of J-, H-, and K-band observations? Finally, I recommend to describe the wavelength ranges covered by the different modes. There will be readers who are not experienced with astronomical filter bands.*

We only use the *J*-, *H*- and *K*- bands as these are the most common observations made by the NOT for determination of colour temperature. We now describe these as "*J*-, *H*- and *K*- bands (with 50% transition levels at 1.165-1.328 nm, 1.484-1.780 nm, and 2.038-2.363 nm, respectively)". We also indicate the fixed "dead-time" between observations of around 10s, leading to a repetition rate of 20.8 seconds between successive observations.

The text now reads: "The detector has a dead-time of about 10 s due to read out processes after each integration." at the end of section 1.

*P.3, L.8: I can understand that only a single spectrum is used for the illustration of the data reduction procedure. However, the discussion in almost the entire paper (except for the outlook) is based on a single series of three 10.8 s exposures (each in a different band). If the whole data reduction and analysis is optimised for these spectra, it is not clear how changes in the observing mode (filter band, spectral resolution, exposure time), observing conditions (atmospheric transmission and turbulence), and the calibration data (with a possible impact on the robustness of the reduction) could*

*affect the quality of e.g. the OH rotational temperatures.*

The data reduction presented here was performed on a variety of spectra from the NOT covering a range of conditions. The single set of spectra shown here represent the shortest integration-times, and thus the most challenging reduction due to signal to noise constraints. This is now made clear in the paper.

The following text is now added: "This exposure time was chosen as an example as it is the shortest exposure time available in the archive, although the methods presented here are developed and optimized for a variety of conditions with integration times up to 600 s, which is the longest integration time available in the archive. "

*P.3, L.9: The given star is probably a telluric standard star, i.e. the corresponding spectra are used to correct atmospheric molecular absorption. Is there a special reason for this selection? In principle, the observations of astronomical science targets or spectrophotometric standard stars could have been used as well.*

The star in this selection of data is not a telluric standard star. These spectra were chosen because of the short integration time (a worst-case scenario; see previous comment) and because standard stars make up a small fraction of the NOT observations. Thus, not using a telluric standard star is more representative of the data within the NOT archive.

*P.3, L.10: Why is the exposure time of the example only 10.8 s? This causes uncertainties in the line measurements and hence the OH rotational temperatures, which could be avoided by longer exposures. Of course, large time differences can reduce the coupling between the OH intensities in the spectra of different filter bands which are taken consecutively. Nevertheless, observing times distinctly longer than 10.8 s would still be well below the Brunt-Väisälä period. Was the selection by purpose (considering the arguments given above) or were there restrictions with respect to the data that was provided by the NOT for this study?*

As stated in the previous two comments, these spectra were chosen from the entire

NOT archive as a worst-case scenario to illustrate the capabilities of the data reduction technique. It also provided sequential filters covering the range of near infrared OH bands. Longer exposure times would, of course, provide higher quality data that could be used to address a variety of scientific questions. However, since the purpose of this paper is to provide a description of the steps necessary to recover atmospheric parameters from typical point-source astronomical data we felt it important to choose the most challenging case. The repetition time is indeed much less than the Brunt-Väisälä period (or even half this period as Nyquist limitations dictate), but the short integration times and high spatial resolution does allow the study of high-frequency wave instabilities using such instrumentation. The results of the science questions addressed with the instrument will be presented in subsequent works in journals focused less on measurement techniques.

*P.3, L.31: "A Gaussian function with a width of 0.31 nm": What was used for the J- and K-band observations? For constant resolving power, the line width is proportional to the wavelength. Moreover, the resolving power could/should depend on the observing mode*
The NOTCam can operate with two resolving powers R=2500 and R 5500. However, the bulk of the *J-*, *H-* and *K-* band observations in the NOT database are made with a resolving power of R=2500. The reduction takes place in "pixel-space" rather than wavelength space, and the monochromatic line shape, the image of the front slit projected on the detector, is determined by the camera optics of the spectrograph and is not a function of the dispersion. Of course, lambda doubling and wavelength dependence of the optics is an issue. We now specify in the text that the line shape determined in pixels was found not to vary across the wavelengths used, and represents a line width of 0.42, 0.73 and 0.87 nm at the centres of the *J-*, *H-* and *K-* bands, respectively.
The text now reads: "Gaussian functions with a full width half maximum of 0.42, 0.73 and 0.87 nm were found to fit the line shape of the resulting OH lines in the centres of the J-, H- and K-bands, respectively, to within the noise present in the data. The fitting

however is performed in units of pixels rather than nm. The line width is constant in pixel units for each band separately at 2.27 pixels."

*P.4, L.8 and Fig. 2: The reduced spectrum does not seem to be flux calibrated. Fig. 2 shows ADU as intensity units. It can lead to a significant bias if such a spectrum is used for the derivation of OH rotational temperatures. Flat-fielding is only for the correction of small-scale variations. For the slowly varying intensity variations and the absolute calibration, instrument response curves derived from observations and reference tables of spectrophotometric standard stars are required. Only the effect of atmospheric extinction by scattering can usually be neglected in the near-IR.*

As stated in the text, the spectra are in photon flux units. Figure 2 has now been modified to show this. Although the data shown does not have an absolute photon-radiance or photon-irradiance calibration, the relative calibration has been performed. For the calibration, the colour temperature of the lamp used to perform the flat fielding was used to correct for the relative instrument response. Unfortunately, the original submission included data that had been processed without this correction. Data without this corrections appeared 1 to 4 K too warm. The data and figures have been corrected. We now also state explicitly in the text when describing Figure 2 that the calibration is only relative, and mention that an absolute calibration could be achieved if appropriate standard stars are used.

The text now reads: "The spectrum has been corrected using a relative spectral radiance calibration derived from the flat-field source. However, an absolute calibration could be achieved using observations of a standard star of known intensity. The absolute calibration is however not needed for this work as we are only interested in the hydroxyl temperatures."

*P.4, L.11: How were the signal-to-noise ratios estimated? Poisson noise?*
As stated in the text, the standard deviation of the line's signal across the spatial dimension of the slit was used as an estimate of noise for each line. The average signal was then used to determine the signal-to-noise ratio.

*P.4, L.17: The continuum underlying the OH emission is removed by applying a Butter-worth filter. This is not the most obvious way to separate line from continuum emission. Were therefore also other approaches tested? For example, continuum windows can be defined on both sides of the interesting lines. Afterwards, the continuum at the line positions can be interpolated and subtracted.*

The use of a high-pass filter to remove low "spectral-frequency" continua has been demonstrated in the literature (e.g. (Espy et al., 1997)and references therein). The technique of subtracting a line whose slope is determined by the continuum on either side of a line is, in fact, the same as applying a high-pass filter (Kennedy, 1980; Owens, 1978). The difference, however, is that the shape and cut-off of this effective filter are dependent upon the data used to construct it; in this case the values of the continuum on either side of the spectral-line determine the slope of the continuum under the line, and hence filter's shape and cut-off frequency in Fourier space. Since this is applied separately to each spectral line, the line-shape of the spectral line will likely vary across the spectrum, requiring the effect of the filter on the line shape of the synthetic spec-trum to be determined for each individual line. Thus, we have chosen a technique that applies a uniform filter to both the data and synthetic spectra, affecting each line in the spectrum consistently. This allows fitting the filtered version of the synthetic spectrum to a filtered version of the data in a least-squares sense.

*P.4, L.19 and Fig. 3: It is stated that the Butterworth filter should have a "minimal im-pact" on the OH lines. However, Fig. 3 reveals that there are major changes in the line shape. Strong negative residuals occur. Do this complex features reflect the true line intensities in a reliable way? This could be tested using alternative approaches (see comment on P.4, L.17).*

This is poorly expressed in the paper. The filter is chosen to have minimal impact on the temperatures derived from the OH. The process used was to fit the tempera-tures using filters with different cut-off values. The initial fit yielded a high temperature that rapidly dropped to a plateau value for increasing values of cut-off frequency. At very high values, the derived temperatures fell as the filter reduced the intensity of the

smaller, high-N lines to the noise level. The value of the cut-off frequency used was in the centre of the plateau region to ensure that changes in the continuum or the noise level did not affect the derived temperatures.

This has now been explained in the text, which now reads "This filter was optimised through repeated tests on synthetic OH spectra to have minimal impact on the temperatures derived from the OH rotational lines."

*P.4, L.20: The resulting spectrum is normalised to 1. However, it varies around 0. Hence, is this approach sufficiently robust?*

The text has been clarified and now reads: "After filtering, the spectrum was normalized such that the sum over the absolute values of the spectrum was equal to one."

*P.4, L.22: Fitting a model spectrum instead of direct line measurements can cause relatively large systematic errors if the model assumptions in terms of airglow physics and the adaption to the observed data (e.g. line-spread function) are not sufficiently accurate. Was this approach chosen because of the relatively low spectral resolution?*

The spectral fitting technique used has been validated against individual line measurements using an exponential fit. Since the line-shape function reproduces the instrumental function to within the noise in the data, the approach of fitting a synthetic spectrum is equivalent.

*P.4, L.22: "known OH line strengths": The Einstein-A coefficients for OH lines are not that well known. Ratios of Einstein-A coefficients for lines from different OH bands can easily vary by a factor of 2 or more depending on the set used. This study appears to be based on those from the HITRAN database, i.e. Goldman et al. (1998). Have the authors ever considered to also use OH molecular parameters from other publications, e.g. van der Loo & Groenenboom (2007, 2008) or Brooke et al. (2016) (to list relatively recent papers). Since this study combines P-, Q-, and R-branch data, the choice of the coefficients can be critical for comparisons of rotational temperatures from OH bands with different line sets (more critical than in the case of P-branch lines only).*

The term "known OH line strengths" has been replaced with "tabulated OH line

strengths" along with the Rothman et al. (2013) reference. The comparison of temperature for different line strengths, while an important scientific endeavour, is beyond the scope of this paper. However, it is interesting to note that as one goes from the Mies to the HITRAN line strengths, the use of the P-, Q- and R- branches together reduces the impact of the line strengths on the fitted temperature. This is due to the difference in line strengths being in the opposite direction for the P- and R-branches, and nearly zero for the Q-branch

*P.4, L.23: The assumption of a Boltzmann distribution with a single temperature is a strong simplification. In fact, any change of the line set will affect the resulting temperature (see Noll et al. 2015). Apart from uncertainties in the Einstein-A coefficients, this is caused by line-specific amounts of non-LTE contributions. They vary depending on vibrational level, rotational level, electronic substate, and observing time. Any OH rotational temperature is a pseudo-temperature (deviating from the kinetic temperature). Whether this is critical for a scientific study is a different question. In any case, one should take OH rotational temperatures and related comparisons with care.*
We agree. Here we have followed the work of Pendleton et al. (1993), itself based upon the work of Polanyi and Woodall (1972) that indicates thermalization of levels whose rotational energy-level spacing is less than kT. In the Pendleton et al. work, this generally occurs for N less than or equal to 4 for vibrational levels up to 7, and levels above 4 were not used here to minimize the effect. Again, an analysis of the non-thermalization of OH is a study which could be undertaken using data from the NOT, but is beyond the scope of this paper focused on the measurement technique. Thus, we fitted a common temperature to the distribution as has been done in e.g. Espy and Hammond (1995).

*P.4, L.28: The success of the model fit also depends on the accuracy of the instrumental line shape. Are the model spectra calculated using the simple Gaussian with fixed width mentioned at L. 31 of P. 3? Have the authors checked how the results on the OH rotational temperatures depend on the line-spread function (within the estimated uncertainties)?*

The line shape has been shown to reproduce the instrumental line shape to within the signal to noise of the instrument across the spectral range used in this work. It has also been shown to reproduce the line shape in spectra with much longer integration times. Small mismatches in the line shape, along with any wavelength positional errors, will increase the residuals and result in a larger fitting error being reported. As stated earlier, the fitting takes place by converting wavelengths to pixels, where the line-shape is constant.

This is made clear in the text in section 2 where it now reads: "The fitting however is performed in units of pixels rather than nm. The line width is constant in pixel units for each band separately at 2.27 pixels."

*P.5, L.1: "higher levels may not be thermalized": This is also likely for lower levels (see comment on P.4, L.23).*
Again, an interesting point that will be addressed in subsequent investigations using the NOT. Here we are following the work of Pendleton et al. (1993)

*P.5, L.3: Is "accurate" the correct word to describe the quality of the OH rotational temperatures? As discussed before, the systematic errors related to the resulting temperatures can be relatively large. Uncertainties in the Einstein-A coefficients, non-LTE effects, line-spread function, and also atmospheric absorption (see comment on P.5, L.32) can be critical.*
This has been changed in the text, which now reads: "The temperature fitting routine was tested against model data with added Gaussian white noise where temperatures were varied between 150 and 400 K. The resulting fit was found to reproduce the input rotational temperatures to within the fitting errors of the temperature parameter."

*P.5, L.5: An error of 0.56 K is remarkably small. How was it calculated?*
The fitting error was determined by calculating the $\chi^2$ distribution and changing the parameter value until $\chi^2$ changed by 1, as outlined in Bevington (2003) on page 147. The fitting error can also be calculated using the variance of the residuals multiplied by the square of the diagonal element of the covariance matrix for the fitted parameter.

[Figure]

The uncertainty in the parameter is then taken as the square root of this product. The two methods agree.

*P.5, L.6: "may not be completely Gaussian": This is not unexpected. For example, the Gaussian might be convolved with a boxcar, which considers the influence of the entrance slit.*
As mentioned in other comments, the Gaussian line-shape used fits the lines over the spectral range presented here to within the precision of the measurements. The same was true of longer integration spectra.
For clarification, the text has been changed to: "The residual spectrum (data - fit) is shown in Fig. 3(d), where it may be seen that although small differences in the wavelength calibration or line shape may occur, the fitted spectrum accurately represents the observation."

*P.5, L.7: "the model used is robust": Note that this statement applies to the spectrum that was used to optimise the analysis. For a different observation, this might not be true anymore (even if the unconsidered systematic model errors are neglected).*
This sentence was intended to indicate that the model was robust over the different OH bands used in this study. The paper has been modified to indicate this and that other spectra taken under different conditions yielded similar results.
The text now reads: "This and similar tests for the other Meinel bands and different integration times demonstrate that the model used is robust [. . .]"

*P.5, L.11: Fig. 4 shows relatively large errors of the wavelength calibration. In part, the shifts are higher than a line width. In particular, the spectrum of OH(7-4) does not fit well. Another example is the mismatch of the highest P-branch lines of OH(8-6) and OH(9-7). What is the influence of these wavelength calibration errors on the model fit and the corresponding rotational temperatures?*
I believe that the reviewer has misinterpreted Figure 4, which does not have a wavelength scale. The data are on a relative wavenumber scale that has been set to zero at each band centre. Since Bv is a strong function of vibrational level for OH, the different

P lines will not line up exactly. The purpose of the figure is to show the portions of each band that are available for fitting, with the line designation listed. This is now clarified in the description of the figure.

The text of the figure caption now reads: "Spectra from the *H*-, *J*- and *K*-bands in wavenumber units on a relative scale. The Q(1) lines of each spectrum are aligned and the vibrational transitions are labeled."

*P.5, L.17: "longer integrations ... would enhance the data quality": The authors assume that the 10.8 s exposure in the J band is too short for reliable temperature measurements. Why was this not tested using a sufficiently long exposure? Only in this way, it is possible to distinguish statistical from systematic errors.*

As stated, the purpose of the paper was to display the ability to recover OH airglow information from the NOTCam under challenging conditions. As many of the NOTCam spectra in the archive are at this shortest integration time, we chose this.

Text changed to: "Due to the lower intensities of these $\Delta v=3$ bands, the signal to noise is substantially reduced for this short integration which is typical of the NOTCam data. As a result, the fitting error for these bands was very large. While longer exposures, or the co-addition of sequential short exposures, would improve the fitting error of the OH (7,4) band, the R and Q branches of the (8,5) band are overlapped by the optically thick $O_2$ Infrared Atmospheric band at 1270 nm, and the filter cut-off reduces its $P_1(4)$ line to near the noise level. Thus, the temperatures for the (8,5) bands would remain compromised even for longer integrations."

*P.5, L.20: Assuming a fixed separation for layers of adjacent vibrational levels of 0.5 km can only be a very rough estimate since this difference is highly variable and also depends on which vibrational levels are compared (e.g., von Savigny et al. 2012; Xu et al. 2012). Hence, Fig. 4 essentially shows OH rotational temperatures as a function of vibrational level, even if the ordinate provides altitudes in kilometres. It would be good to clarify that the resulting temperature gradients are only qualitatively correct.*

*P.5, L.25: "the altitude [sic] ... were assigned": The fixed altitude step size of 0.5 km*

*could be mentioned here again.*

The peak concentrations of the neighbouring vibrational levels are, on average, separated in altitude by 0.5 km (von Savigny et al., 2012). Even though the absolute peak altitudes are known to vary with season (Gao et al., 2010), they can be taken as constant on the time scales of a few minutes over which this experiment was executed. With a steady-state model using a neutral atmosphere from the Naval Research Laboratory Mass Spectrometer and Incoherent Scatter Radar Empirical model (NRLMSISE) (Picone et al., 2002), the altitude of the (9,7) transition was fixed and the relative altitudes of the other vibrational transitions were assigned using the altitude separation of 0.5 km, the average separation found by von Savigny et al. (2012).

*P.5, L.26: The term "atmospheric temperature profile" can be misleading. Apart from the already mentioned uncertainty in the true emission peak altitude (comment on P.5, L.20), it also has to be considered that this profile is strongly smoothed due to the typical emission layer widths of 8 to 9 km. The most critical issue is the fact that the given rotational temperatures are a combination of kinetic temperatures and non-LTE effects. However, the term "atmospheric temperature profile" suggests that a kinetic temperature profile is shown.*

Indeed, the text is changed to reflect the uncertainties and now reads: "provide an estimate of temperature gradients present". In addition, the reference to Schubert et al, 1990, is added.

*P.5, L.27: For a comparison of the NOTCam data and the NRLMSISE model, the latter should be smoothed considering a typical OH emission profile. This should significantly reduce the model temperature gradient (see Noll et al. 2016).*

The temperature profile has now been smoothed using a typical profile calculated using MSIS and a steady state model for the average emission rate profile of an OH Meinel band. Figure 5 now shows both gradients from MSIS, with and without this smoothing. The effect of this smoothing is a slight steepening of the temperature gradient, but the data still fits this model gradient to within two sigma.

*P.5, L.28: In view of all the effects which were not considered for the temperature comparison, I do not think that a safe statement on possible similarities can be made. In principle, there should not be an agreement between the observed and the modelled data due to the contributions of non-LTE effects to the former.*

While any non-LTE effects would still be present, we feel that the corrected statement above is justified and reflects the agreement between the kinetic model temperature and the derived OH temperatures, notwithstanding any non-LTE effects. The use of different OH bands to infer temperature gradients is not new. In fact, the Schubert et al (1990) work shows that temperature changes ascribed to non-LTE effects may be effects of large gradients in the atmosphere, which can far exceed climatological values due to wave effects (Xu et al, 2000).

This section now reads: "The rotational temperatures derived from each of the individual vibrational bands then provide an estimate of temperature gradients present similar to what has been done by Perminov et al. (2007) and Schubert et al. (1990).

Fig. 5 shows the distribution in altitude for the data presented here together with the NRLMSISE model kinetic temperature for the corresponding location and time. In addition, the NRLMSISE profile has been smoothed with the volume emission rate profile of an OH band derived from the steady-state OH model to reflect the effect of the OH layer width on the rotational temperatures. The temperature profile from this model is consistent with the gradients estimated from our data for the (3,1), (4,2), (5,3), (6,4) and (9,7) transitions. It is important to note that the *J*-, *H*- and *K*-band data shown in Fig. 5 were acquired using sequential 10.8 s observations, spanning only 6 minutes. Thus, large deviations from the climatological gradients can occur due to wave activity (Xu et al., 2000). While the agreement here may be fortuitous, a time sequence of short integrations may give insight into the wave-induced temperature gradients. While the (8,6) transition is anomalously high, it was found that the P(2) and P(4) lines are partially absorbed by atmospheric $H_2O$ and CO (Jones et al., 2013; Noll et al., 2012)."

*P.5, L.29: It would be better to show the temperatures from the OH(7-4) and OH(8-5) bands. Otherwise one could think that the authors want to hide something. Even if the*

*quality of the rotational temperature measurements for these two bands are lower, they can be plotted if realistic error bars are assumed. There would not be an issue with the signal-to-noise ratio of these observations if longer exposures were taken for this paper (see also comment on P.5, L.17).*

The temperatures from the OH(7-4) and OH(8-5) bands are now shown in figure 5. The figure is reproduced as figure 1 in the attachments as it now appears in the text.

*P.5, L.30: In view of all the effects which were not considered for the temperature comparison, it cannot be stated that the temperature for OH(8-6) is "anomalously high". Moreover, it is even expected that the values for the 8th vibrational level are the highest (Cosby & Slanger 2007; Noll et al. 2015, 2016).*

*P.5, L.31: "P(2) and P(4)": The numbers for the electronic substates are not indicated here. What was exactly done to identify that these lines are partially absorbed? The given references suggest that the Cerro Paranal sky model could have been used for this.*

*P.5, L.32: How were the correction factors for the line intensity decrease in the model estimated? In principle, one needs a very high resolution atmospheric transmission spectrum (resolving power > 10Ë€6) for realistic atmospheric conditions. In particular, the amount of water vapour is highly variable. Therefore, it is important to estimate this amount if selected lines are affected. Apart from data from global weather models, direct measurements using the observed data are most promising. The telluric absorption in the observed stellar spectrum can be investigated with suitable fitting tools. Fortunately, the P-branch lines in the OH(8-6) band are mostly affected $CO_2$, which is much less variable than water vapour. Nevertheless, the high resolution transmission spectrum and a model of the airglow emission line profiles are needed to derive the correction factors. There are several papers on this topic (e.g., Espy & Hammond 1995; Noll et al. 2015; Chadney et al. 2017). Patrick Espy is even a co-author of this paper.*

*P.6, L.1: A change of the temperature by 8% is a huge effect, which can cause large uncertainties depending on the quality of the line absorption derivation. More details*

*are needed to evaluate the uncertainties in the resulting OH rotational temperatures.*
*P.6, L.2: It is not clear what "same technique" means since the procedure for the line*
*absorption correction is not described (see also comment on P.5, L.32).*

As these five comments (P.5, L.30; L.31; L.32 and P.6 L.1; L.2) all refer to the same issue, they have been taken together. The Cerro Paranal sky model was used and the resulting spectrum used to reduce the lines whose positions were taken from the HITRAN database. A more complete description of the procedure is now given in the text (see below).

As per the 8% difference, this is what comes from fitting a synthetic spectrum with and without the atmospheric transmission applied. Since the same line strengths are used in preparing the synthetic spectrum as were used to fit it, the line strengths themselves should play a limited role. Given that correcting the barely noticeable absorption within the (7,4) and (3,1) bands analysed by Espy and Hammond (1995) results in a 2% to 3% change in temperature, it is not unreasonable that correction of the (8,6) band results in a change of 8%.

The paragraph now reads: "While the (8,6) transition is anomalously high, it was found that the P1(2) and P1(4) lines are partially absorbed by atmospheric $H_2O$ and CO (Jones et al., 2013; Noll et al., 2012). To model the impact of this atmospheric absorption, synthetic spectra with rotational temperatures between 130 K and 300 K were created. When these were fitted with the technique described above, this input temperature could be retrieved. When these synthetic spectra were however first multiplied by a high resolution (0.002nm) absorption spectrum for seasonally averaged conditions obtained from the Cerro Paranal sky model (Jones et al., 2013; Noll et al., 2012), the fitted temperature is approximately 8% higher than the original synthesized temperature. This would account for the higher fitted temperature of the observed (8,6) band shown in Fig. 5. Using this same technique, the temperatures for all other vibrational-rotational transitions presented in Fig. 5 were examined and found not to be significantly affected by atmospheric absorption."

*P.6, L.4: This statement could change (see also comment on P.5, L.28).*

*P.6, L.5: The authors provide a long list of papers which apparently support their conclusion of decreasing OH rotational temperatures with increasing vibrational level. However, Oberheide et al. (2006), French & Mulligan (2010), and Dyrland et al. (2010) only provide results for a single OH band, which cannot be used for this investigation. In the case of Lübken et al. (1990), the results for OH(8-3) and OH(3-1) are quite similar. However, they are based on two independent instruments with different measurement periods. Sivjee & Hamwey (1987), Phillips et al. (2004), and Wrasse et al. (2004) compare rotational temperatures from the OH(8-3) and OH(6-2) bands. The first two studies find higher temperatures for OH(8-3), which is in agreement with the uncorrected NOTCam results but disagrees with the authors' conclusions. Espy & Hammond (1995) find similiar temperatures for OH(7-4) and OH(3-1), which also does not support the conclusions. Only the northern winter measurements used by Perminov et al. (2007) appear to be in good agreement.*

*P.6, L.8: The list of publications finding an increase of the rotational temperatures with vibrational level could be complemented by Cosby & Slanger (2007). This investigation was the first one which clearly showed this remarkable temperature dependence with a maximum for the 8th vibrational level, which can only be explained by a significant contribution of non-thermalised rotational populations.*

We are merely reporting the results we have. Once again, the intent of this paper is not to take issue with the results that have indicated NLTE within the OH band temperatures. The data sample presented is far too short to test that result. Rather we wished to show that the data and technique presented here produce believable results that, in this case, reflect the kinetic temperature of the atmosphere. To show that this result is not an outlier, we have quoted a number of publications showing OH temperatures that reflect the kinetic temperature measured by SABER (Dyrland et al., 2010; French and Mulligan, 2010; Innis et al., 2001; Oberheide et al., 2006). Similarly, our result showing only a slight change of temperatures with vibrational level is at odds with the NLTE results presented by Cosby and Slanger as well as Noll. To show that this is not unusual, we have quoted results showing measurements of high and low vibrational

bands where the temperatures are the same (to within two sigma), or very similar:
Same to within 2 sigma: (Espy and Hammond, 1995; Lübken et al., 1990; Perminov et al., 2007)

Similar to within 2 sigma: (Phillips, 2004; Sivjee and Hamwey, 1987; Wrasse et al., 2004)

For completeness, we should point out that the Wrasse et al. (2004) results show a decrease in temperature with vibrational level in agreement with our results. The Sivjee and Hamwey (1987) results see a significant difference in the (8,3) and (6,2) temperatures only after removing the effects of atmospheric tides. Even then the temperature difference is $6 \pm 4$ K (2 sigma), a difference which they attribute to different emitting altitudes in a positive temperature gradient of $5 \pm 3$ K/km. The Phillips et al. results, which do not remove the effects of atmospheric tides, show a difference of $3.9 \pm 0.8$ K (1 sigma), although for no-moon and clear sky conditions this drops to $2.6 \pm 1.2$K.

Again, our point was not to make a comment on the presence or absence of NLTE. Our measurements span too short of a time and could reflect atmospheric temperature gradients caused by tides and waves (see response to comment P5 L28). Rather we wanted to show that the results of our analysis on the NOTCam data are within the realm of what has been observed before. In an effort to qualify that agreement, we attempted to point out that Noll et al. as well as Cosby and Slanger (dropping this reference was a regrettable oversight on our part and we thank the reviewer for calling this to our attention) have shown increasing temperatures with vibrational level which they attribute to NLTE.

The text now reads: "With the intent of demonstrating that the NOTCam provides atmospheric data that can be used to supplement other astronomical data sets used for aeronomic studies (e.g Osterbrock et al. (1996), Noll et al. (2015), Cosby and Slanger (2007)), a sample, short-integration spectrum has been analysed, and both the analysis procedure and results presented. With the exception of the weak or compromised v'=7 and 8 levels discussed earlier, the temperature variation with vibrational level observed using the NOTCam and analysed here reflects the MSIS kinetic temperature

to within two sigma. This is consistent with previous observations using ground-based spectrometers or interferometers (e.g.Innis et al. (2001); Oberheide et al. (2006); French and Mulligan (2010); and Dyrland et al. (2010)). However, it must be pointed out that the NOTCam measurements cover a very short time span and may be affected by gravity-wave perturbations of the climatological temperature gradient of the atmosphere represented by MSIS. The temperature distribution with vibrational level observed here shows a small decrease toward higher levels, although this decrease is, apart from the v'=9, not significant at the two sigma level. This is at odds with the measurements of Noll et al. (2015) and Cosby and Slanger (2007), who show strong increases in temperature with vibrational level that they attribute to non-thermodynamic equilibrium effects. However, our results are consistent with the results of Lübken et al. (1990), Espy and Hammond (1995), Wrasse et al. (2004), and Perminov et al. (2007), who show similar or decreasing temperatures with increasing vibrational level. Once again, the short duration of this demonstration data is not able to resolve this apparent discrepancy, but the larger NOTCam data set may prove useful in this regard."

*P.6, L.12: NOTCam competes with other instruments at the NOT. Nevertheless, an observing time fraction of 15% is relatively small. Moreover, it has to be considered that NOTCam is also an imaging instrument and that there are various spectroscopic modes with different wavelength coverages. Therefore, the number of spectra that cover a given OH band should be relatively small. This means that it is unlikely that a good annual coverage is achieved by the scheduled observations. It appears to be necessary to combine data from the whole period between 2007 and 2016 to perform variability studies. Exceptions could be individual observing nights with a good time coverage. However, this depends on the design of the corresponding astronomical project if archival data are used. Would it possible to perform dedicated airglow observations? I would like to see a more detailed discussion of the data set in terms of such limitations.*

"Would it be possible to perform dedicated airglow observations?" The NOT is a community instrument and will entertain proposals. Please see the NOT website for details

and the new figure 7.

*P.6, L.13 and Fig.6: The figure lacks information on the total number of spectra for a certain day of year and in which years these observations were performed. One could use more symbols, symbol sizes, and/or colours to provide more details. The figure only shows data taken between November 2007 and June 2016. Were there no suitable observations before this period although the spectroscopy mode started in August 2003 according to P. 2, L. 29?*

*P.6, L.18: Apart from showing the histograms in Fig. 7, it would be helpful if the total numbers of spectra for the different bands are provided in the text. Since the paper discusses a set of consecutive observations in three bands, it would also be interesting to know how many combined observations of this kind exist in the data archive. As the exposure time seems to be an issue at least in the J band, it would be good to learn more about the distribution of exposure times.*

Once again, these two comments are taken together. The purpose of the figures is to show the distribution of measurements on a seasonal and nightly basis to demonstrate any bias in the observations that may preclude some studies. Many experiments can be conceived using the data, but each would have its own requirements. For example, some might want a long integration time in order to perform spectroscopic studies. Nights with repeated, short integration times, even when there are other observations interspersed, could be used by adding the individual exposures. However, other studies might want to study time evolution throughout the night, and the long cadence time due to interspersed, non-OH observations, may create too long of a delay between observations. Still other experiments may want to study the vibrational development of the OH, and therefore want interspersed observations of the J, H and K bands. It is beyond the scope of this paper, perhaps any paper, to provide a comprehensive summary of the data in the archive that would be useful for every conceivable experiment. We have now provided a link to the NOT searchable archive in the paper the data from which will enable the interested reader to conceive their own experiments, and added histogram plots to figure 7 showing the hours of data available in the archive for each

hour of the night. The attached figure 2 below is the new figure 7 from the paper.
The reason for the time span between 2007 and 2016 is only the availability in the archive. To aid the referee and reader in the design of any particular experiment, the link to the archive is now stated in the acknowledgements. The text now reads: "Archive data from the NOTCam can be accessed via http://www.not.iac.es/observing/forms/fitsarchive/"

*P.6, L.20: I am not sure whether this rather limited data set can be called "extensive".*
The word "extensive" has been dropped.

*P.6, L.31: Does the NOTCam archive already include suitable time series with high temporal resolution or would it be necessary to apply for observing time for a dedicated study?*
Rather than a referee comment on the manuscript, this sounds much more like a question requesting information for a particular experiment that would be better addressed by examining the archive (see response above).

*P.7, L.3: In the case of observations of astronomical objects, the spatial resolution is effectively reduced by the moving telescope. This issue should be discussed. Would it be possible to perform NOTCam observations without telescope tracking? However, if this was possible, there would be possible issues related to the data reduction of 2D spectra with various star traces.*
We had assumed the change in angular resolution would be obvious when tracking stars. We now distinguish between "staring resolution" (4 arc-min × 0.6 arc-sec) and a "worst-case tracking resolution" (6.7 × 2.7 arc-min) at the shortest integration times. The latter, of course, depends upon the orientation of the slit relative to the scanning direction and the integration time.
The text now reads: "Here the observations have a spatial resolution on the airglow layer of about 100m (which is 4 arc minutes), sampled with approximately 1000 pixels in the spatial direction of the NOT detector. This resolution is achieved with the telescope in "staring mode", which means without star tracking. The resolution with star

tracking is dependent on the slit orientation and is in worst case up to $6.7 \times 2.7$ arc minutes over a 10.8 s integration."

For astronomical use, the tracking mode is standard. However, if one were to perform observations in a staring mode, one would sensibly orientate the slit perpendicular to the motion of the stars. Thus, a star would cross the cross-slit field of view in 0.04 seconds. Depending upon the relative intensity of the star, this may produce a noticeable spectral background that would, depending upon the spectral structure of the object, be removed by the filtering described in the data processing outlined here. Again, a poorly designed experiment may have a very bright star, planet or the moon crossing the field of view that would adversely affect the observation. We do, however, feel that the design of a staring experiment as to orientation, where to look, or what backgrounds may occur, should be the responsibility of the individual experimenter.

However we have added the text: "While staring observations would recover the full spatial resolution of the instrument, care should be taken with the slit orientation and choice of direction to ensure that bright astronomical objects do not interfere with the airglow observations."

Technical corrections:
*P.5, L.25: altitude -> altitudes (see comment on P.5, L.25)*
*P.6, L.5: It would be better to write "northern winter" (or something similar) instead of only "winter".*
*P.6, L.8: measurement -> measurements*
*P.14, Fig.5: In the coordinates for La Palma, the "N" for northern is missing.*
*P.7, L.2: "large spatial resolution" -> "low spatial resolution"*
These technical corrections are incorporated into the text.

[revised manuscript text omitted]

---

## Author Response (AR2)

We thank the referee for the suggestions for revision. We will answer the points in detail in the following:

*Abstract: Despite the major changes in the text, the abstract is still the same (except for a single word). Please check carefully whether the current abstract still agrees with the rest of the paper. For example, note that the reported error of about 0.5 K is smaller than all uncertainties in Fig. 5. The best accuracy seems to be 0.7 K, which was achieved for OH(5,3). The uncertainties for the other bands appear to be significantly larger.*

The fitting error mentioned was indeed a remnant of the previous version and has been corrected. The rest of the abstract still reflects the content and intent of the paper and therefore remains unchanged.

The text now reads: "Rotational temperatures representative of the background atmospheric temperature near 90 km, the mesosphere and lower thermosphere region, can be fitted to the OH rotational lines with a precision of around 0.7 K."

*P. 3, L. 13 and P. 5, L. 28: In the response to my comments, it was stated that the observed astronomical object is not a (telluric) standard star. However, the confusing term is still used in the paper at two positions. In astronomy, standard stars are only observed for data calibration and processing.*

This is true and has been changed in these two occasions. On P. 4 L. 22, we use "standard star" in its astronomical meaning, and therefore this one instance is left unchanged.

The text now reads: "To demonstrate the data reduction procedures required to optimise the OH signal from routine astronomical observations, a single *H*-band spectroscopic exposure toward the star, 21 Vir (spectral type B9V, H = 5.64 mag), with an exposure time of 10.8s was used."

And: "Fig. 4 shows an overview of all the measured OH vibrational transitions recorded in the *H*-, *J*- and *K*-band spectra taken toward the same star, extracted in the manner outlined above".

*P. 5, L. 21 and P. 14, L. 2 (Fig. 3): The rotational temperature for the OH(5,3) band has changed in Sect. 3.1 but not in the caption of Fig. 3. This discrepancy has to be corrected. If I interpret the response letter correctly, the change is related to a modified approach for the relative flux calibration.*

This is true. We apologize for the confusion and have changed the temperature in the caption of figure 3 to the correct value.

The text now reads there: "The fitted rotational temperature is 186.5 ± 0.7 K."

*P. 6, L. 27: The most critical molecule for the absorption of OH(8,6) lines in the lower atmosphere is CO2. There are no CO lines in the relevant wavelength regime.*

This is of course true and the missing index was added.

The test now reads: "While the (8,6) transition is anomalously high, it was found that the P(2) and P(4) lines are partially absorbed by atmospheric $H_2O$ and $CO_2$"

*P. 14, Fig. 4: The caption of this figure has improved. However, it could be even better if it was directly stated that the labelled tickmarks only provide rough positions of the corresponding lines.*

[revised manuscript text omitted]